

# Evaluation of TROPOMI operational standard NO₂ column retrievals (from version 1.3 to 2.4) with OMNO2 and QA4ECV OMI observations over China

Jianbin Gu[1,2], Xiaoxia Liang[3], Shipeng Song[4], Yanfang Tian[5], Liangfu Chen[1,2], Jinhua Tao[1,2,*]

[1]State Key Laboratory of Remote Sensing Science, Aerospace Information Research Institute, Chinese Academy of Sciences, Beijing, 100101, China

[2]University of the Chinese Academy of Sciences, Beijing, 100049, China

[3]College of Resource Environment and Tourism, Capital Normal University, Beijing, 100089, China

[4]College of Earth Sciences, Guilin University of Technology, Guilin, 541004, China

[5]State Key Laboratory of Environmental Criteria and Risk Assessment, Chinese Research Academy of Environmental Sciences, Beijing, 100012, China

*Corresponding author: Jinhua Tao

Address: 20 Datun Road, Chaoyang District, State Key Laboratory of Remote Sensing Science, Aerospace Information Research Institute, Chinese Academy of Sciences, Beijing, 100101, China; Tel: +86-10-64889545; Fax: +86-10-64889545; E-mail: 254293974@qq.com



**Abstract.**
The TROPOMI satellite instrument plays a key role in nitrogen dioxide ($NO_2$) monitoring on account of its
unprecedented spatial resolution and stable quality of data. However, since 2019, TROPOMI operational $NO_2$
retrieval has improved and updated in three versions (1.4, 2.2 and 2.4), with significant impact on retrieved $NO_2$
column. Thus, studies including both TROPOMI $NO_2$ data before and after the activation of these versions could
show artificial jumps. Moreover, up to date evaluation result of TROPOMI $NO_2$ data in current version 2.4 is not
yet well documented in the literature. Therefore, in this work, we focus on evaluating TROPOMI's capability to
detect $NO_2$ under the different retrieval version conditions, by comparing with OMNO2 data and QA4ECV OMI
data over China. We find a 38 % increase of tropospheric $NO_2$ in version 1.4 due to improved FRESCO-wide
cloud retrieval, and a 14 % increase in version 2.2 due to adjusted surface albedo for cloud-free scenes. We show
that the upgrade to version 2.4 with new DLER surface albedo, led to an increase by $3 \times 10^{14}$ molecules $cm^{-2}$ of
tropospheric $NO_2$ over vegetation. Furthermore, we demonstrate that TROPOMI data shows strongest
tropospheric $NO_2$ seasonal variation compared to OMNO2 data and QA4ECV OMI data, and this seasonal effect
was enhanced with the tropospheric $NO_2$ retrieval version upgrades. Additionally, we examine for the first time
the change of TROPOMI AMFs (air mass factors) in the different versions, and based on it, we arrive at a
correction for the underestimation of TROPOMI $NO_2$ column in previous versions. We also find a 33 %
overestimation of $NO_2$ reduction during the COVID-19 lockdown over China when using TROPOMI data before
and after the activation of the $NO_2$ version 1.4.
**Keywords:** $NO_2$, TROPOMI, OMI, QA4ECV, evaluation.



## 1 Introduction

Nitrogen dioxide ($NO_2$) is an important pollutant trace gas as a primary pollutant and as a precursor to ozone and
fine particulate matter production (Cooper et al., 2022). Thus, fast, efficient and accurate monitoring of ambient
$NO_2$ from regional to global scale is indispensable for air quality evaluation and atmosphere pollution control.
Among methods of $NO_2$ monitoring, satellite remote sensing has been widely applied with its large-scale, real-
time, simultaneous and high-frequency dynamic monitoring mode. Since 1997, a series of studies on $NO_2$
monitoring with satellite instruments such as Global Ozone Monitoring Experiment (GOME), Global Ozone
Monitoring Experiment-2 (GOME-2), SCanning Imaging Absorption spectro Meter for Atmospheric
CHartographY (SCIAMACHY), Ozone Monitoring Instrument (OMI), and TROPOspheric Monitoring
Instrument (TROPOMI) has been made, which include monitoring of $NO_2$ variations (Van der A et al., 2006;
Schneider et al., 2015), $NO_2$ transport phenomena (Nowlan et al., 2014), evaluation of nitrate deposition (Liu et
al., 2017b), estimation of nitrogen oxides ($NOx = NO + NO_2$) emission amounts (Curier et al., 2014; Park et al.,
2021) and inference of surface $NO_2$ concentrations (Gu et al., 2017; Kim et al., 2021).

Among the above-mentioned satellite instruments, the OMI instrument launched in July 2004 is the Dutch-
Finnish contribution to National Aeronautics and Space Administration (NASA)'s Earth Observing System (EOS)
Aura sensor. It has been used widely to conduct research by applying its long-term observations of $NO_2$, due to
its high spatial resolution and daily global coverage (Levelt et al., 2018; Liu et al., 2017a). As a successor of
OMI, since launched in October 2017, the European TROPOMI satellite sensor which is on board the Sentinel-
5-Precursor (S5P) has began to play an important role in $NO_2$ monitoring on account of its unprecedented spatial
resolution and stable quality of data (van der A et al., 2020; Ding et al., 2020; Griffin et al., 2019; Zhao et al.,
2020). Up to date TROPOMI and OMI are the main data sources in satellite observation of $NO_2$ (Biswal et al.,
2021). Moreover, previous studies focusing on comparative assessment of TROPOMI and OMI $NO_2$ data have



been conducted (Van Geffen et al., 2020; Riess et al., 2022), and the results suggest that data quality of
TROPOMI NO$_2$ observations is significantly improved (Griffin et al., 2019; Wang et al., 2020).

However, currently the following issues should be noted when making study by using TROPOMI NO$_2$ data.
Firstly, TROPOMI NO$_2$ retrieval algorithm has improved and updated several versions, and three of them
(version 1.4, 2.2 and 2.4) have significant impact on retrieved NO$_2$ column productions. For instance, Riess et al.
(2022) found that the improved NO$_2$ retrieval algorithm in version 1.4 led to increases of TROPOMI NO$_2$
columns of up to 40 % as compared to version 1.2 in Europe. Thus, studies including both TROPOMI NO$_2$ data
before and after the activation of these versions may show artificial jumps. Secondly, the changes in TROPOMI
NO$_2$ columns caused by these version updates are different, due to their different aspects in improvements of
NO$_2$ retrievals. Additionally, up to date evaluation result of TROPOMI NO$_2$ data in current version 2.4 (since
July 2022) is not yet well documented in the literature. Therefore, in this work, we focus on evaluating
TROPOMI's capability to detect NO$_2$ column in its retrieval version 1.3-2.4 over China, and measuring changes
caused by the activation of these versions.

In previous studies NO$_2$ observations released by ground-based remote sensing techniques such as Pandora and
multi-axis differential optical absorption spectroscopy (MAX-DOAS) instruments are generally used to compare
and assess NO$_2$ retrievals derived with satellite instruments (Compernolle et al., 2020; Griffin et al., 2019).
However, systematic and consistent ground-based NO$_2$ observation data has been only provided till November
2017     (e.g.     QA4ECV     MAX-DOAS     data     sets,     available     at     http://uv-
vis.aeronomie.be/groundbased/QA4ECV_MAXDOAS, last access: 9 October 2022). Thus, in this work, we
report a comprehensive evaluation of TROPOMI NO$_2$ version 1.3-2.4 data products with OMNO$_2$ version 4.0
data products from November 2019 to September 2022 over China. Moreover, the differences between



TROPOMI and OMI NO$_2$ standard data are not only from their instrumental differences, but also from a total
uncertainty on their algorithmic differences. Therefore, besides the OMNO$_2$ data, QA4ECV (Quality Assurance
for Essential Climate Variables) OMI NO$_2$ version 1.1 data, which follows a more similar retrieval algorithm as
TROPOMI NO$_2$ data, is also used to compare.

Our study is structured as follows. Sect. 2 provides an introduction to TROPOMI and OMI instrument, as well as
their retrievals of NO$_2$ column, including three main TROPOMI NO$_2$ retrieval version updates. The information
of QA4ECV OMI NO$_2$ measurement is also given in this section. Sect. 3 presents NO$_2$ columns, NO$_2$ spatial-
temporal distributions, and seasonal variations over China derived with the TROPOMI data in the different
versions (1.3, 1.4, 2.2 and 2.4), by applying the OMNO2 data and QA4ECV OMI NO$_2$ data as references. The
differences between TROPOMI and OMI NO$_2$ measurements are analysed in relation to their tropospheric NO$_2$
column discrepancies. Potential causes of the differences (e.g. surface albedo error, cloud parameters and priori
profile shape uncertainty) are then discussed. Moreover, AMFs (air mass factors) in the different TROPOMI
NO$_2$ retrieval versions are obtained, and the overestimation of NO$_2$ reduction during COVID-19 lockdown over
China caused by using TROPOMI data before and after the version 1.4 is adjusted. Finally, a conclusion is given
in Sect. 4.

**2 Description of the data sets**
**2.1 S5P TROPOMI NO$_2$**
TROPOMI is a nadir-viewing spectrometer aboard European Space Agency (ESA)'s the S5P satellite (Van
Geffen et al., 2020). It is designed to monitoring atmospheric components including ozone (O$_3$), NO$_2$, sulfur



dioxide ($SO_2$), carbon monoxide (CO) and formaldehyde (HCHO) with daily global coverage, as the successor
of OMI (Veefkind et al., 2012). TROPOMI traces a sun-synchronous polar orbit with an equator crossing at
about 13:30 local time, and provides observation data in four channels covering ultraviolet (UV) to shortwave
infrared wavelengths. In the visible (VIS) channel (400 nm-496 nm) used for $NO_2$ retrieval, TROPOMI's
horizontal resolution at true nadir is improved to an unprecedented extent than the previous satellite instruments
(De Smedt et al., 2018). Its observation individual pixels are 7 km (5.5 km since August 2019) as an integration
time of 1.08 s in the along-track, and 3.5 km in the across-track direction at the middle of the swath. Along the
across-track direction there are 450 ground pixels in a row, and these pixel sizes remain more or less constant
towards the edges of the swath (the largest pixels are 14 km wide) (Van Geffen et al., 2020).

The TROPOMI $NO_2$ retrieval which is developed by the Royal Netherlands Meteorological Institute (KNMI)
(Van Geffen et al., 2020) consists of a three-step procedure: (1) Deriving of a total atmospheric $NO_2$ slant
column density (SCD) using the Differential Optical Absorption Spectroscopy (DOAS) retrieval method in the
405 nm-465 nm spectral range. (2) Separation of the retrieval total $NO_2$ SCD into a stratospheric $NO_2$ SCD and a
tropospheric $NO_2$ SCD based on the TM5-MP model (Williams et al., 2017). (3) Normalization of a tropospheric
$NO_2$ vertical column density (VCD) from the retrieval tropospheric $NO_2$ SCD by applying an appropriate AMF.
The AMF is defined as the ratio of the observed SCD of the absorbing trace gas along the slant optical path from
sun to satellite, and the vertical column density above the point at the surface area the satellite is viewing. More
details of the TROPOMI $NO_2$ retrieval are described in the product Algorithm Theoretical Basis Document (Van
Geffen et al., 2020).



### 2.1.1 Improved FRESCO-wide cloud retrievals in version 1.4

Tropospheric AMF uncertainty is the largest source of satellite-derived tropospheric $NO_2$ column uncertainty for polluted scenes, ranges between 20 %-50 %, leading to a total uncertainty in tropospheric $NO_2$ column in the 30 %-60 % range (Liu et al., 2021). The $NO_2$ AMF to harmonize the conversion of SCD into VCD is calculated using the Doubling-Adding KNMI (DAK) radiative transfer model (Lorente et al., 2017), and the input parameters to the TROPOMI $NO_2$ AMF calculation are surface albedo climatology (Kleipool et al., 2008), priori $NO_2$ profiles (Williams et al., 2017), viewing geometry (satellite and solar angles), terrain height and cloud parameters (Riess et al., 2022), including cloud pressure retrieved with the TROPOMI FRESCO cloud algorithm (driven by the 761 and 765 nm $O_2$ absorption depth). With the introduction of version 1.4 in December 2020, a new FRESCO-wide cloud algorithm was introduced and implemented in the TROPOMI operational $NO_2$ retrieval to address the high-bias in the previous FRESCO cloud pressures used in version 1.0-1.3. The main improvement by the FRESCO-wide algorithm is an overall reduction of the observed cloud pressures, resulting in a decrease of AMFs and a substantial increase of $NO_2$ in the retrievals in polluted regions.

### 2.1.2 Adjusted surface albedo in version 2.2-2.3

From July 2021 onwards, for TROPOMI $NO_2$ version 2.2, a surface albedo adjustment was implemented to avoid negative cloud fractions while maintaining radiance closure. For instance, cloud fraction varies between 0 and 1 on physical grounds, and when the actual surface albedo is lower than expected from the Kleipool et al. (2008) surface albedo climatology, it leads to a negative cloud fraction. In the previous TROPOMI $NO_2$ version retrievals, this was clipped to 0. But with the implementation of version 2.2, surface albedo is decreased to match cloud fraction equal 0, and thus, ensure radiance closure (Van Geffen et al., 2022). Additionally, the Kleipool et al. (2008) surface albedo climatology based on OMI data does not cover the near-infrared wavelengths in use by



the FRESCO algorithm to derive cloud properties, and, instead, up to version 2.3 the surface albedo database
used by the FRESCO algorithm is based on GOME-2 observations (Tilstra et al., 2017) at 758 and 772 nm. The
overpass time of GOME-2 is several hours earlier relative to OMI and TROPOMI, which is in favour of the
Kleipool surface albedo climatology for the $NO_2$ retrieval, and to determine the cloud fraction in the $NO_2$
window (S5P-KNMI-L2-0005-RP, available at https://sentinel.esa.int/documents/247904/2476257/Sentinel-5P-
TROPOMI-ATBD-NO2-data-products; last access: 9 October 2022). As a consequent, these lead to a significant
increase (10 %-15 %) of TROPOMI tropospheric $NO_2$ for cloud-free scenes on top of the increase for pixels
with small cloud fractions in version 1.4 related to the improved FRESCO-wide cloud retrievals.

We note that the $NO_2$ fit window of the wavelength was not correct in TROPOMI version 2.2, with negligible
effect on the $NO_2$ column retrieval, while it was corrected in version 2.3 (405 nm-465 nm). Thus, TROPOMI
$NO_2$ version 2.3 product is the most complete and consistent to date. Zhang et al. (2023) report on the
improvement of TROPOMI version 2.3 $NO_2$ columns, and their impact on emission estimates specifically over
China in times of COVID-19 lockdowns. Additionally, to consistent with the extents of improvements for
version 2.2 and 2.3 released by the ESA S5P/TROPOMI $NO_2$ algorithm change record (S5P-MPC-KNMI-PRF-
NO2, available at http://sentinels.copernicus.eu/web/sentinel/technical-guides/sentinel-5p/products-algorithms/;
last access: 9 October 2022), in this work TROPOMI version 2.2 and 2.3 $NO_2$ column products are collectively
referred to as the former.

**2.1.3 Alternative surface albedo climatology in version 2.4**
With the introducing of TROPOMI $NO_2$ version 2.4 in July 2022, a Directional Lambertian Equivalent
Reflectivity (DLER) climatology derived from TROPOMI observations replaced the original surface albedo



climatologies derived from OMI and GOME-2 in older versions 1.0-2.2. This new DLER climatology is applied
in cloud fraction and cloud pressure retrievals in the NO$_2$ window, using in the TROPOMI NO$_2$ AMF calculation.
It has several advantages in the represent of the directionality or viewing-angle dependence of the scattering at
the surface, as well as the improved spatial resolution of the surface albedo climatology database from 0.5 x 0.5
degree to 0.125 x 0.125 degree. But up to date the impact of the DLER climatology in version 2.4 to the
TROPOMI NO$_2$ column retrieval has not yet been released.

### 179 2.2 Aura OMI NO$_2$

The OMI sensor launched in July 2004 was installed on NASA's Earth Observing System Aura satellite. It is
designed to continue the Total Ozone Mapping Spectrometer (TOMS) record for O$_3$ and other atmospheric
component products such as NO$_2$, SO$_2$ and HCHO (Boersma et al., 2007). OMI is a nadir-viewing imaging
spectrograph that measures direct and atmosphere-backscattered sunlight within an UV - VIS range of 270 nm-
500 nm (Levelt et al., 2006). It traces a sun-synchronous ascending polar orbit with an equator crossing time of
13:30. The spatial resolution of OMI NO$_2$ product is about 13 x 24 km$^2$ at nadir. Along the cross track, OMI
pixel sizes vary with viewing zenith angles from 24 km in the nadir to approximately 128 km in extreme viewing
angles of 57 degree along the edges of the swath (Boersma et al., 2007). Since October 2004 OMI has provided
various trace gas concentration observations with daily global coverage. It should be noted that after May 2008
with the introduction of the row anomaly, OMI no longer provides daily global coverage.

The OMI NO$_2$ (OMNO2) retrieval algorithm consists of a three-step procedure. (1) a spectral fitting algorithm to
calculate total NO$_2$ SCD in the 402 nm-465 nm spectral range. (2) a stratosphere-troposphere separation scheme



to derive tropospheric and stratospheric $NO_2$ VCDs. (3) determination of AMF to convert SCD to VCD. Detailed
descriptions of the OMNO2 retrieval algorithm were provided by Bucsela et al. (2013) and Celarier et al. (2008).

Up to date a series of significant conceptual and technical improvements in the OMNO2 retrieval has been made.
A new scheme for seperating stratospheric and tropospheric components was implemented in the OMNO2
version 2.1 (Lamsal et al., 2014). With the introduction of version 3.0, a significant advance of $NO_2$ SCD
retrieval was developed (Krotkov et al., 2017). The current version, 4.0, a several changes for improved $NO_2$
AMF and VCD calculations are introduced, including applying a new geometry dependent surface Lambertian
Equivalent Reflectivity product in $NO_2$ retrieval (Fasnacht et al., 2019), as well as improved cloud parameter
retrievals (effective cloud fraction and optical centroid pressure from a new cloud OMCDO2N algorithm)
(Vasilkov et al., 2018).

**2.3 QA4ECV OMI NO$_2$**
The EU Seventh Framework Programme QA4ECV project (http://www.qa4ecv.eu, last access: 9 October 2022)
was initiated in 2014. It aims to demonstrated how reliable and traceable quality information can be provided for
satellite and ground-based measurements of climate and air quality parameters (Compernolle et al., 2020). The
project developed and applied a quality assurance framework on new and improved satellite data records of the
atmosphere ECVs including $NO_2$, HCHO and CO.

The QA4ECV OMI $NO_2$ version 1.1 product is retrieved from OMI Level 1 UV-Vis spectral measurements, and
its retrieval algorithm is based on the DOAS approach, like the OMNO2 product (Boersma et al., 2018). The
differences between OMNO2 and QA4ECV OMI $NO_2$ product account for estimating the stratospheric SCD and



calculating the tropospheric AMF (e.g. the prior $NO_2$ profiles information and the cloud retrieval). Previous
evaluations suggest that the discrepancies between OMNO2 and QA4ECV OMI tropospheric $NO_2$ VCD
retrievals typically lead to small but spatially widespread differences of up to 0.5-1 x $10^{15}$ molecules $cm^{-2}$
(Compernolle et al., 2020). On the other hand, the retrieval of tropospheric $NO_2$ from QA4ECV OMI proceeds
along the same lines as from TROPOMI, and is thus similar in many aspects (Riess et al., 2022).

**2.4 Screening criteria**
The details on the retrievals of the TROPOMI, OMNO2 and QA4ECV OMI tropospheric $NO_2$ column are given,
see Table 1. In this work we compared the TROPOMI tropospheric $NO_2$ version 1.3-2.4 data to the OMNO2
tropospheric $NO_2$ version 4.0 data and QA4ECV OMI tropospheric $NO_2$ version 1.1 data in order to evaluate
their capabilities to detect $NO_2$. We selected these satellite data for tropospheric $NO_2$ column evaluation if the
following conditions are met:
(1) TROPOMI $NO_2$ column products taken a sufficient quality of retrieval (qa_value > 0.50);
(2) $OMNO_2$ column products where the XtrackQualityFlags field is equal to 0, for selecting only rows which
have not been affected by the row anomaly;
(3) QA4ECV OMI $NO_2$ column products where the processing_error_flag field is equal to 0;
(4) All satellite $NO_2$ column products taken an effective cloud fraction less than 0.2;
(5) All satellite $NO_2$ column products taken a satellite solar zenith angle less than 80 degree.

**Table 1**. Retrievals for the TROPOMI version 1.3-2.4, OMNO2 version 4.0 and QA4ECV OMI version 1.1
tropospheric $NO_2$ column used in this study.

| TROPOMI | TROPOMI | TROPOMI | TROPOMI | OMNO2 | QA4ECV |
| --- | --- | --- | --- | --- | --- |



| | v1.3 | v1.4 | v2.2 | v2.4 | v4.0 | OMI v1.1 |
|---|---|---|---|---|---|---|
| Public data period | 20 Mar 2019- 29 Nov 2020 | 29 Nov 2020-01 Jul 2021 | 01 Jul 2021- 17 Jul 2022 | 17 Jul 2022- | 01 Oct 2004- | 01 Oct 2004- 30 Mar 2021 |
| Spectral fitting | Van Geffen et al. (2020) | Van Geffen et al. (2020) | Van Geffen et al. (2020) | Van Geffen et al. (2020) | Marchenko et al. (2015) | Zara et al. (2018) |
| Surface albedo | Kleipool et al. (2008) 5-year climatology at 0.5° x 0.5° | Kleipool et al. (2008) 5-year climatology at 0.5° x 0.5° | Kleipool et al. (2008) 5-year climatology at 0.5° x 0.5° (adjusted) | TROPOMI DLER climatology at 0.125° x 0.125° | Kleipool et al. (2008) 5-year climatology at 0.5° x 0.5° | Kleipool et al. (2008) 5-year climatology at 0.5° x 0.5° |
| A priori $NO_2$ profiles | Daily TM5-MP at 1° x 1° | Daily TM5-MP at 1° x 1° | Daily TM5-MP at 1° x 1° | Daily TM5-MP at 1° x 1° | Monthly Global Modelling Initiative data at 1° x 1.25° | Daily TM5-MP at 1° x 1° |
| Clouds retrieval | FRESCO | FRESCO-wide | FRESCO-wide | FRESCO-wide | OMCDO2N | OMCLDO2 |
| Stratospheric correction | Data assimilation in TM5-MP | Data assimilation in TM5-MP | Data assimilation in TM5-MP | Data assimilation in TM5-MP | Bucsela et al. (2013) | Data assimilation in TM5-MP |




## 3 Results and discussion

### 3.1 NO₂ columns and trends

We start with evaluating TROPOMI's capability to detect tropospheric NO$_2$ with the OMNO2 and QA4ECV
OMI NO$_2$ observations. First, we create 7 x 7 km$^2$ TROPOMI tropospheric NO$_2$ VCD version 1.3-2.4 daily data
and 0.25 x 0.25 degree OMNO2 tropospheric VCD version 4.0 daily data from November 2019 to September
2022, as well as QA4ECV OMI tropospheric NO$_2$ VCD version 1.1 daily data from November 2019 to March
2021, as described in Section 2.4. Then, we derive the daily means of these data sets over China which have not
been selected for co-sampling, in order to ensure their respective data validity, and the monthly means of relative
differences between them are further calculated (Fig. 1). Meteorological effects were generally minor at the
national scale.

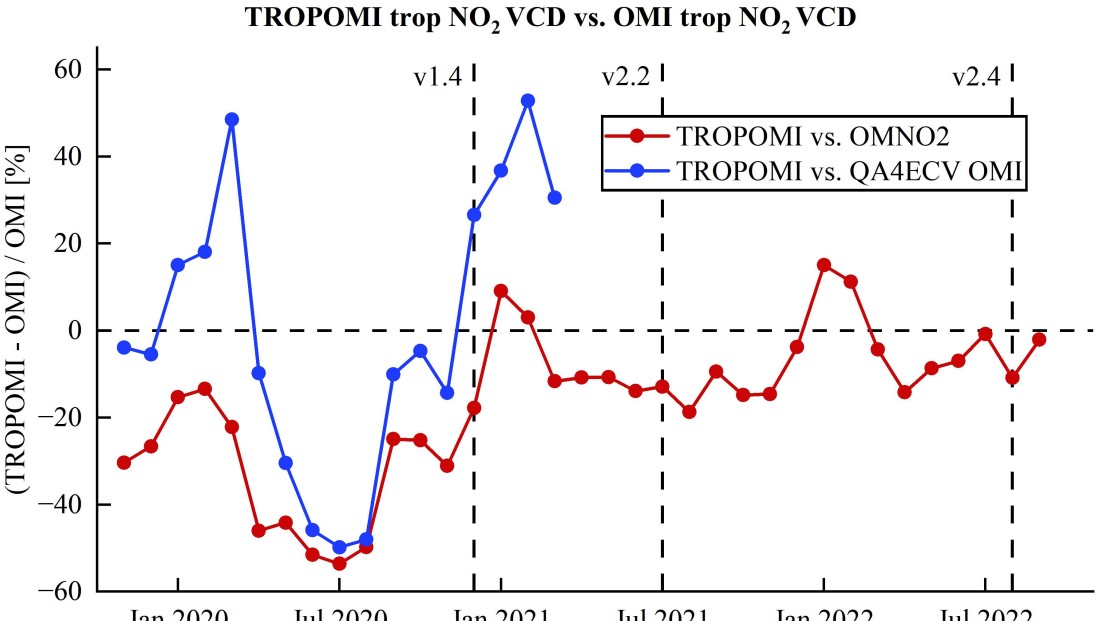

**Figure 1.** Relative differences between TROPOMI (version 1.3-2.4) and OMNO2 tropospheric NO$_2$ VCDs from November 2019 to September 2022 (red), and between TROPOMI and QA4ECV OMI tropospheric NO$_2$ VCDs from November 2019 to March 2021 (blue) over the whole China. The black vertical line represents the date when the TROPOMI NO$_2$ retrieval version started.

The tropospheric NO$_2$ VCDs over China derived from TROPOMI version 1.3 observations are overall lower by 33 ± 14 % and 11 ± 28 % than those derived from OMNO2 and QA4ECV OMI observations respectively (Fig. 1). This can be explained by the overestimation of the FRESCO cloud pressures, and subsequently the overestimation of the AMFs, and thus, the underestimation of the tropospheric NO$_2$ columns for scenes with small cloud fractions in the TROPOMI NO$_2$ version 1.3. Moreover, the TROPOMI tropospheric NO$_2$ VCDs have the largest decrease in the summer months (e.g. 52 % for June, 54 % for July and 50 % for August), and the smallest decrease in the winter months (e.g. 15 % for January, 13 % for February and 22 % for March), as



compared to the OMNO2 tropospheric VCDs. Similar seasonal differences exist in the comparison of the
TROPOMI tropospheric $NO_2$ VCDs to the QA4ECV OMI tropospheric $NO_2$ VCDs (e.g. −46 % for June, −50 %
for July, − 48 % for August and 15 % for January, 18 % for February, 49 % for March). These seasonal
differences in decrease of the TROPOMI tropospheric $NO_2$ version 1.3 columns relative to the tropospheric $NO_2$
columns derived from OMI exhibit a summer maximum and winter minimum, in contrast to the winter
maximum and summer minimum in TROPOMI or OMI total $NO_2$ columns.

We also compare the TROPOMI tropospheric $NO_2$ VCD daily data from December 2020 to June 2021 (the
entire version 1.4 period), to the OMNO2 and QA4ECV OMI tropospheric $NO_2$ VCD daily data over China. We
find that the extent of the decrease between the TROPOMI and OMNO2 tropospheric $NO_2$ VCDs has become
smaller between December 2020 and June 2021 ($1.89 \pm 3.08 \times 10^{14}$ molecules cm$^{-2}$) than between December
2019 and June 2020 ($6.59 \pm 3.18 \times 10^{14}$ molecules cm$^{-2}$). Similarly, the extent of the increase between
TROPOMI and QA4ECV OMI tropospheric $NO_2$ VCDs has become larger between December 2020 and March
2021 ($6.99 \pm 1.74 \times 10^{14}$ molecules cm$^{-2}$) than between December 2019 and March 2020 ($2.13 \pm 2.92 \times 10^{14}$
molecules cm$^{-2}$). Therefore, we conclude that the upgrade to version 1.4 with the improved FRESCO-wide cloud
retrieval, led to a significant increase (about $5 \times 10^{14}$ molecules cm$^{-2}$) of tropospheric $NO_2$ columns as compared
with the previous version. As a consequence, the low bias in the TROPOMI tropospheric $NO_2$ columns prior to
November 2020 was (at least partly) addressed.

An increase (22 %-35 %) in the TROPOMI tropospheric $NO_2$ version 1.4 columns over China is measured by
comparing with the QA4ECV OMI tropospheric $NO_2$ columns between December 2020 to March 2021 than
between December 2019 to March 2020 (Fig. 2). Similar increase (19 %-32 %) was observed by comparing the
TROPOMI version 1.4 tropospheric $NO_2$ VCDs to the OMNO2 tropospheric VCDs during the same periods. We



conclude that the TROPOMI NO₂ column enhancement (of up to 38 %) was identified from version 1.3 to 1.4
over China, due to the improved cloud information retrievals. This conclusion is in agreement with previous
validation studies by Riess et al. (2022) who found that the improved cloud pressures in version 1.4 led to
increases of TROPOMI NO₂ columns of up to 40 % in Europe, and by (S5P-MPC-KNMI-PRF-NO2) who found
an increase of up to 50 % in TROPOMI NO₂ version 1.4 over East Asia.

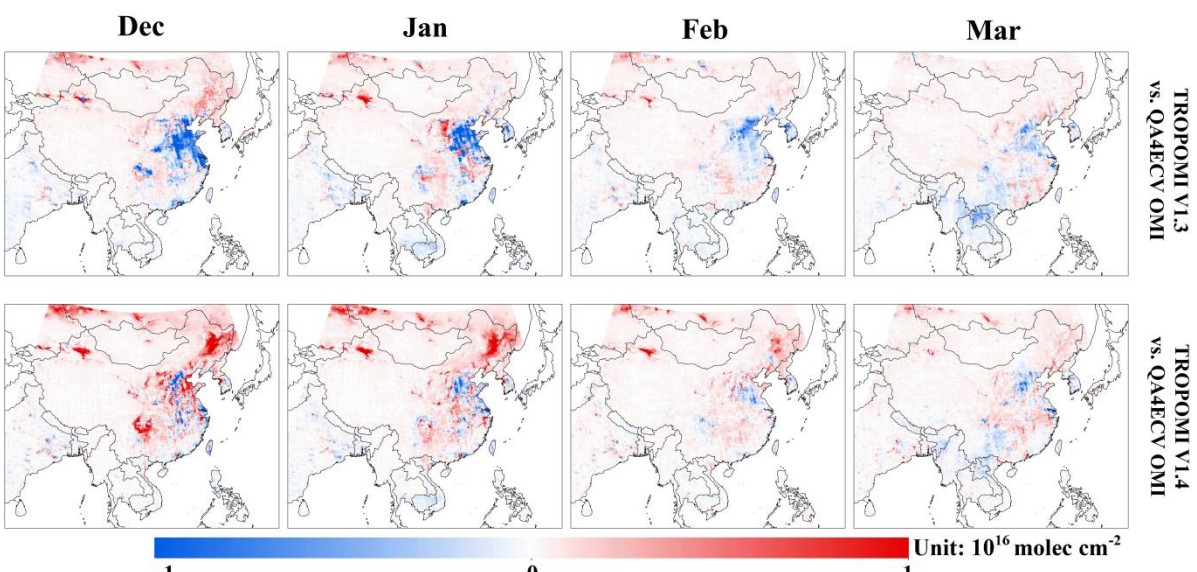


**Figure 2.** Differences in monthly mean tropospheric NO₂ columns derived from TROPOMI data and QA4ECV
OMI data (TROPOMI minus QA4ECV) between December 2019 and March 2020 (first row, TROPOMI version
1.3), and between December 2020 and March 2021 (second row, TROPOMI version 1.4). NO₂ columns derived
using TROPOMI observations gridded at 0.25 x 0.25 degree resolution.

Since the QA4ECV OMI NO₂ data product is available before 30 March 2021, here, we compare the TROPOMI
NO₂ columns only to the OMNO₂ columns after the date. Throughout the entire version 2.2 period (from July



2021 to June 2022), the TROPOMI tropospheric $NO_2$ VCDs are lower by $1.30 \pm 2.52 \times 10^{14}$ molecules $cm^{-2}$
compared to the OMNO2 tropospheric VCDs over China. Furthermore, this decrease is weakest in the winter of
2021/2022 ($-1.84 \times 10^{14}$ molecules $cm^{-2}$) and strongest in the summer of 2021 ($2.57 \times 10^{14}$ molecules $cm^{-2}$). This
seasonal trend of the difference between the TROPOMI tropospheric version 2.2 $NO_2$ and the OMNO2 is similar
with that between the TROPOMI tropospheric version 1.4 $NO_2$ and the OMNO2. It can be explained by the
surface albedo adjusted to avoid negative cloud fractions while maintaining radiance closure in TROPOMI $NO_2$
version 2.2, and thus, this adjust can lead to a significant increase of tropospheric $NO_2$ columns for cloud-free
scenes which occur frequently in winter and rarely in summer in China. Additionally, an increase of up to 14 %
in the TROPOMI tropospheric $NO_2$ columns in version 2.2 is measured by comparing with the OMNO2
tropospheric columns between December 2021 to March 2022 compared to the previous year.

We compare the TROPOMI tropospheric $NO_2$ version 2.4 daily VCDs retrieved with the new DLER surface
albedo climatology used in the FRESCO-wide cloud fraction and cloud pressure retrievals, to the OMNO2
tropospheric daily VCDs from August to September 2022. As a result, the extent of the difference between the
TROPOMI and OMNO2 tropospheric VCDs has decreased between August and September 2022 ($1.04 \times 10^{14}$
molecules $cm^{-2}$) compared to the previous year ($2.31 \times 10^{14}$ molecules $cm^{-2}$) over China. We find that the DLER
surface albedo climatology in tropospheric AMF calculating in version 2.4 led to a 6 % increase of TROPOMI
tropospheric $NO_2$ columns over China. This is consistent with previous validation study by (S5P-MPC-KNMI-
PRF-NO2) who suggest that the impact of the DLER surface albedo climatology in TROPOMI $NO_2$ version 2.4
retrievals over Europe, North America and East China is relatively minor.

Overall, tropospheric $NO_2$ columns derived from TROPOMI, OMNO2 and QA4ECV OMI provide a similar
initial baseline over China (Fig. 3). They exhibit a clear spatial pattern of tropospheric $NO_2$ with the higher



pollution levels over the Beijing-Tianjin-Hebei (BTH), Yangtze River Delta (YRD) and Pearl River Delta (PRD)
region, due to the combined effects of local developed industrialization and huge population density. On the
other hand, in general, compared to OMNO2, QA4ECV OMI $NO_2$ follows a more similar retrieval algorithm as
TROPOMI $NO_2$, thus comparison between TROPOMI $NO_2$ and QA4ECV OMI $NO_2$ is much more direct as the
algorithmic differences between them will cancel, exposing better the main instrumental differences.
Consequently, the difference between TROPOMI version 1.3 $NO_2$ VCD and OMNO2 VCD (7.18 x $10^{14}$
molecules cm$^{-2}$, 47 %) is considerably larger than that between TROPOMI version 1.3 $NO_2$ VCD and QA4ECV
OMI $NO_2$ VCD (1.97 x $10^{14}$ molecules cm$^{-2}$, 13 %). Additionally, the annual average tropospheric $NO_2$ VCD
over China derived from TROPOMI version 1.3, OMNO2 and QA4ECV OMI is 1.52 ± 0.63 x $10^{15}$, 2.24 ± 0.72
x $10^{15}$ and 1.71 ± 0.55 x $10^{15}$ molecules cm$^{-2}$ respectively. Compared to the OMNO2 tropospheric VCDs, the
lower QA4ECV OMI tropospheric $NO_2$ VCDs are most likely interpreted with their differences between the
tropospheric AMF calculations (and especially the priori $NO_2$ profiles information, see Table 1) (Goldberg et al.,

333    2017).


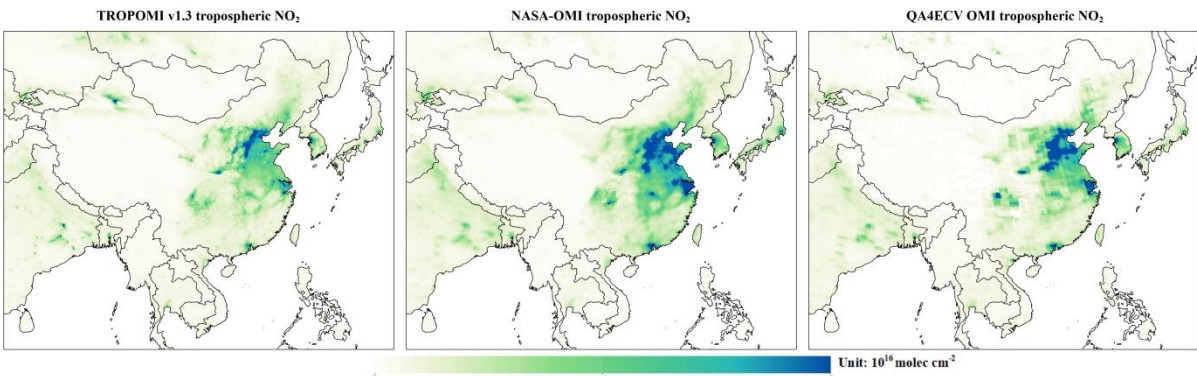


**Figure 3.** November mean tropospheric $NO_2$ columns derived from TROPOMI version 1.3 (left panel), OMNO2
(center panel) and QA4ECV OMI (right panel) observations in 2019.



## 3.2 TROPOMI NO₂ version 2.4 over vegetation

The impact of the upgrade to version 2.4 on TROPOMI NO$_2$ column at national scale are given in Section 3.1.

However, the DLER surface albedo using in TROPOMI version 2.4 accounts for the directionality or viewing-

angle dependence of the scattering at the surface, especially over vegetation in the near infrared. Thus according

to this strong effect of the DLER over vegetation, we evaluate to the new DLER surface albedo in and its impact

on the TROPOMI NO$_2$ columns, to better understand the recent detection of NO$_2$ under condition of vegetation

coverage. In this section, we compare the TROPOMI tropospheric NO$_2$ VCD daily data in August of 2020

(version 1.3), 2021 (version 2.2) and 2022 (version 2.4) over Fujian Province (the province with the highest

vegetation coverage in China), as well as over China as a reference (Fig. 4).

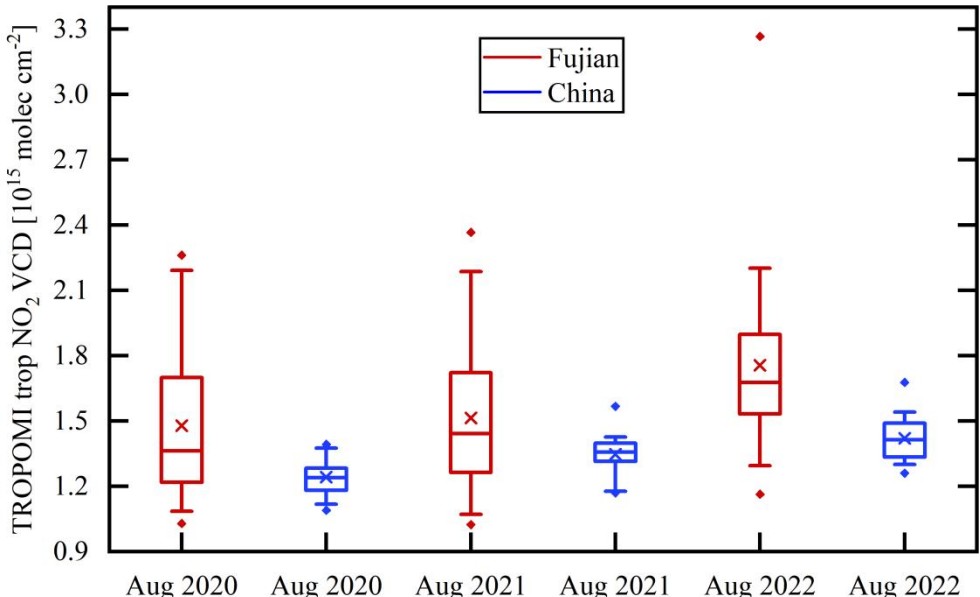

**Figure 4.** Boxplots of daily TROPOMI tropospheric NO$_2$ columns in August of 2020 (version 1.3), 2021

(version 2.2) and 2022 (version 2.4) over Fujian Province (red), and over China (blue). The box edges represent



the 1st and 3th quartiles, the line in the box represents the median, the cross in the box represents the mean, the
dots represent the outlier, and the whiskers represent the 5th and 95th percentiles.

We find that from version 1.3 to 2.2 to 2.4, the TROPOMI tropospheric $NO_2$ column over China is increased by
9 % (1.06 x $10^{14}$ molecules cm$^{-2}$) and 5 % (0.73 x $10^{14}$ molecules cm$^{-2}$) respectively, and in comparison, the
increase in TROPOMI tropospheric $NO_2$ column over Fujian from version 1.3 to 2.2 is relatively minor (2 %,
0.35 x $10^{14}$ molecules cm$^{-2}$), but the tropospheric $NO_2$ enhancements over this region from version 2.2 to 2.4 are
presented, with a substantial increase (16 %, 2.42 x $10^{14}$ molecules cm$^{-2}$). We also compare the TROPOMI
tropospheric $NO_2$ daily VCDs to the OMNO2 tropospheric daily VCDs between August to September in 2021
and 2022 over Fujian. As a result, the upgrade to version 2.4 with the DLER surface albedo, led to a significant
increase (about 3.44 x $10^{14}$ molecules cm$^{-2}$) of tropospheric $NO_2$ columns as compared with the previous version
over vegetation.

**3.3 $NO_2$ seasonal cycle**
$NO_2$ has obvious seasonal variation characteristics with low in summer and high in winter, as $NO_2$ lifetime could
prolong due to low solar irradiances and low specific humidity (Bauwens et al., 2020). Previous studies suggest
that TROPOMI and OMI can effectively reflect the $NO_2$ seasonal variation on account of their high temporal and
spatial resolutions (Dimitropoulou et al., 2020; Meng et al., 2018). Here, we use TROPOMI (version 1.3-2.2),
OMNO2 and QA4ECV OMI $NO_2$ observations from November 2019 to June 2022 to explore their sensitivities
to the $NO_2$ seasonal variation. We select three periods based on the TROPOMI $NO_2$ retrieval version updates as
follows: December 2019 to November 2020 (version 1.3), December 2020 to June 2021 (version 1.4) and July
2021 to June 2022 (version 2.2). Then, we calculate the ratios of the January and June mean tropospheric $NO_2$



VCD in each period to the averaged tropospheric $NO_2$ VCD over the entire period over China retrieved from
TROPOMI, OMNO2 and QA4ECV OMI observations respectively (Fig. 5).

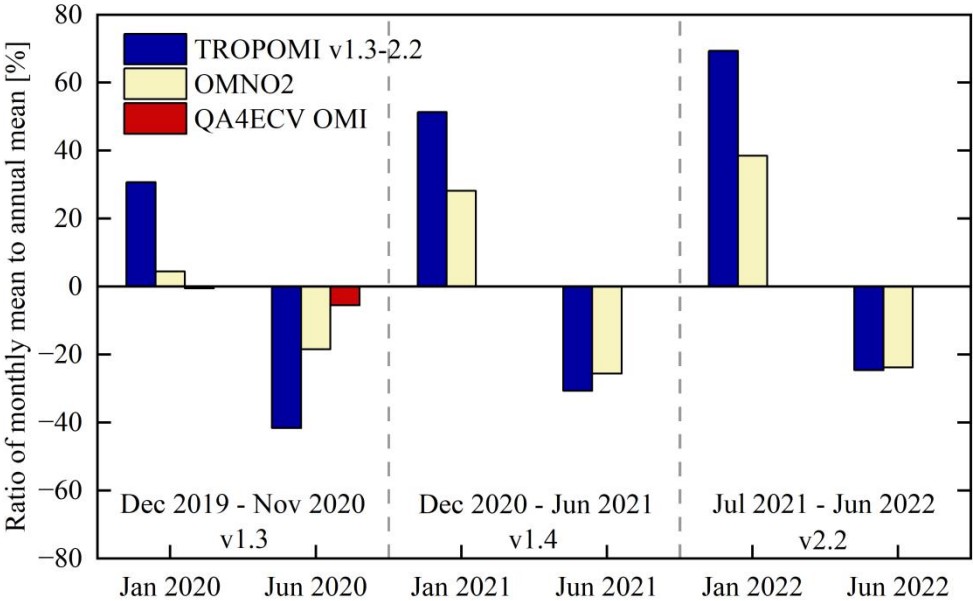

**Figure 5.** Ratios of the January and June mean tropospheric $NO_2$ VCD in each period (December 2019-
November 2020, December 2020-June 2021, July 2021-June 2022) to the averaged tropospheric $NO_2$ VCD over
the period retrieved from TROPOMI, OMNO2 and QA4ECV OMI observations over China.

Overall, TROPOMI data shows strongest seasonal variation of tropospheric $NO_2$ columns compared to OMNO2
data and QA4ECV OMI data. During all three periods, compared to the averages over the entire periods, the
extents of the observed $NO_2$ changes in winter or summer month retrieved from TROPOMI exceed those
retrieved from OMI (Fig. 5). Although QA4ECV OMI follows a more similar $NO_2$ retrieval algorithm to
TROPOMI relative to OMNO2, the increase in winter and decrease in summer of $NO_2$ observed with QA4ECV
OMI (−0.5 % and -5 %) are even smaller than those observed with OMNO2 (4 % and -18 %) over China. Taking



this into account, and the strong seasonal variation of tropospheric NO$_2$ columns present in TROPOMI version
1.3-2.2 data, we conclude that the FRESCO (FRESCO-wide) cloud algorithm using in the NO$_2$ retrieval has a
positive impact on the clear demonstration of seasonal variation of TROPOMI tropospheric NO$_2$.

We find that with the introduction of the FRESCO-wide algorithm in version 1.4 and the adjusted surface albedo
in version 2.2, the ratio of the January mean NO$_2$ column to the averaged NO$_2$ column is increased by 21 % and
39 % in TROPOMI version 1.4 and 2.2 than in version 1.3 over China, respectively. But the ratio for June in
version 1.4 and 2.2 is decreased by 11 and 17 % than in version 1.3. As a consequence, the changes in the
TROPOMI NO$_2$ retrieval version 1.3-2.2 lead to form stronger (weaker) effect of tropospheric NO$_2$ seasonal
variation in winter (summer). This can be explained by the seasonal variation of cloud pressure (Ri et al., 2022),
which is provided more realistic by the FRESCO-wide cloud algorithm in TROPOMI version 1.4, for instance in
the case of low clouds, as well as the adjusted surface albedo for cloud-free scenes in version 2.2, which can
occur more commonly in winter than in summer. Since up to date the TROPOMI NO$_2$ version 2.4 data is
available for only two months, its seasonal variation could be studied in future.

We also create the daily tropospheric NO$_2$ VCDs derived from TROPOMI, OMNO2 and QA4ECV OMI
observations over the BTH, YRD and PRD region in China to compare (Fig. 6). As a result, the monthly means
of tropospheric NO$_2$ VCDs between November 2019 to November 2020 over BTH region are 3.41 ± 0.65
(TROPOMI), 3.01 ± 0.78 (OMNO2) and 4.31 ± 2.24 (QA4ECV OMI) times higher than over China,
respectively. Moreover, these higher trends reached a maximum of 4.37 (in January, TROPOMI), 4.16 (in
January, OMNO2), 8.62 (in January, QA4ECV OMI), and a minimum of 1.94 (in September, TROPOMI), 1.62
(in August, OMNO2), 2.11 (in June, QA4ECV OMI). Similar trends exist over other regions with high pollution
(e.g. YRD and PRD), as demonstrated by Fig. 6. Consequently, these selected pollution regions show more



significant tropospheric NO₂ columns in winter due to anthropogenic emissions. Additionally, we calculate the
differences between TROPOMI and QA4ECV OMI tropospheric NO₂ daily VCDs for each selected pollution
region and each month from November 2019 to November 2020. We see that the ratio of the TROPOMI NO₂
VCD to the QA4ECV OMI NO₂ VCD was closest to 1 in summer months (e.g. 0.93 in July for BTH, 0.97 in
July for YRD, 1.00 in July for PRD), and farthest to 1 in winter months (e.g. 0.55 in February for BTH, 0.71 in
December for YRD, 0.66 in January for PRD). Therefore, compared to the QA4ECV OMI NO₂ retrieval, the
FRESCO cloud algorithm using in the TROPOMI NO₂ retrieval has a strongly positive impact on the
tropospheric NO₂ seasonal cycle, especially in high pollution regions in winter.

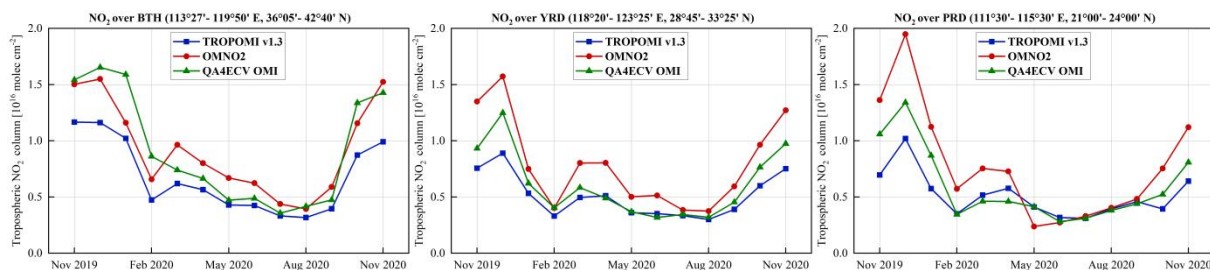

**Figure 6.** Time series of TROPOMI version 1.3, OMNO2 and QA4ECV OMI tropospheric NO₂ columns over
the BTH, YRD, PRD region, which is typical of high NO₂ pollution conditions in China.

**3.4 AMFs and NO₂ column biases**
AMF uncertainties dominate overall satellite-derived NO₂ retrieval errors over polluted regions (Boersma et al.,
2004; Lamsal et al., 2014; Lorente et al., 2017; Martin et al., 2002). For this analysis, we create TROPOMI
tropospheric AMF daily data from January 2020 to August 2022 at a resolution of 7 x 7 km², to study its changes
in the upgrades from version 1.3-2.4 over China (Fig. 7). As a result, the daily AMFs in version 1.4 are lower (of
up to 32 %, on 9 February) than in version 1.3. We find that the AMF reduction from version 1.3-1.4, exhibits a



winter maximum (0.30, 16 % for January and 0.33, 18 % for February) and summer minimum (0.06, 5 % for
May and 0.07, 5 % for June). We conclude that this TROPOMI AMF reduction, range between about 5 % in
summer and 20 % in winter, is mainly due to the implement of the FRESCO-wide algorithm in the operational
$NO_2$ version 1.4. Furthermore, the difference of this reduction in different months is caused by the seasonal
variation of cloud pressure, which is consistent with the seasonal reduction in TROPOMI tropospheric $NO_2$
VCDs from version 1.3 to 1.4, as described in Section 3.3. The TROPOMI tropospheric AMFs in version 1.3-2.4
from 2020 to 2022 over China is given (Table 2).






**Figure 7.** Boxplots of TROPOMI tropospheric daily AMFs in version 1.3 (black), 1.4 (blue), 2.2 (red) and 2.4
(green) from 2020 to 2022 over China.

**Table 2**. TROPOMI tropospheric AMF data in version 1.3-2.4 from January 2020 to August 2022 over China.

| Month | tropospheric AMF (v1.3) | tropospheric AMF v1.4-1.3 difference | tropospheric AMF v2.2-1.3 difference | tropospheric AMF v2.2-1.4 difference | tropospheric AMF v2.4-2.2 difference |
|---|---|---|---|---|---|
| Jan | 1.89 ± 0.12 | -0.30 ± 0.14 | na | -0.09 ± 0.09 | na |
| Feb | 1.78 ± 0.11 | -0.33 ± 0.12 | na | 0.05 ± 0.15 | na |
| Mar | 1.54 ± 0.13 | -0.18 ± 0.14 | na | 0.02 ± 0.12 | na |
| Apr | 1.43 ± 0.10 | -0.13 ± 0.10 | na | 0.03 ± 0.09 | na |
| May | 1.38 ± 0.08 | -0.06 ± 0.09 | na | 0.00 ± 0.09 | na |
| Jun | 1.43 ± 0.03 | -0.07 ± 0.05 | na | 0.01 ± 0.05 | na |
| Jul | 1.43 ± 0.09 | na | -0.06 ± 0.06 | na | na |
| Aug | 1.50 ± 0.06 | na | -0.08 ± 0.07 | na | -0.04 ± 0.07 |
| Sep | 1.49 ± 0.05 | na | -0.07 ± 0.06 | na | na |
| Oct | 1.56 ± 0.06 | na | -0.13 ± 0.10 | na | na |
| Nov | 1.69 ± 0.12 | na | -0.11 ± 0.12 | na | na |
| Dec | na | na | na | 0.01 ± 0.07 | na |


The difference of TROPOMI tropospheric AMF from version 1.4 to 2.2 is relatively minor as compared to that
from version 1.3 to 1.4 (Fig. 7), range between a 5 % overestimation and a 3 % underestimation over China. This
is in agreement with the larger difference of TROPOMI tropospheric $NO_2$ column from version 1.3 to 1.4



relative to that from version 1.4 to 2.2, as described in Section 3.1, reflecting that although the adjusted surface
albedo in version 2.2 could lead to significant decrease of tropospheric AMF, and subsequently increase of
tropospheric $NO_2$ column for cloud-free scenes, but indeed, the extent of the change in tropospheric $NO_2$ column
caused by it is generally smaller than that caused by the improvements of cloud pressure and cloud fraction in
version 1.4 at the national scale. Additionally, we calculate the TROPOMI tropospheric AMF daily data in
August of 2021 (version 2.2) and 2022 (version 2.4) over China, and find that the TROPOMI AMFs in version
2.4 using the DLER climatology are $0.04 \pm 0.07$ lower than in version 2.2.

We also compare the TROPOMI tropospheric AMF version 1.3 daily data to the QA4ECV OMI tropospheric
AMF version 1.1 daily data from January to November 2020 over China. As a result, the TROPOMI AMFs are
higher by $1.57 \pm 0.96$ times compared to the QA4ECV OMI AMFs, mainly due to the differences in their clouds
retrievals using in the tropospheric AMF calculation. Moreover, we find that in all seasons, the TROPOMI
AMFs are higher than the QA4ECV OMI AMFs by factors of $1.42 \pm 0.13$ in January, $0.87 \pm 0.09$ in April, of
$0.64 \pm 0.08$ in July, and $0.72 \pm 0.05$ in October. This change of increment magnitude in different months is
mainly due to the seasonal variation of the cloud pressure using in the TROPOMI $NO_2$ retrieval.

The biases of TROPOMI tropospheric AMFs in different versions presented above have a dominated impact on
the biases of the TROPOMI tropospheric $NO_2$ columns retrieved under these version conditions. Thus, we
calculate the differences of TROPOMI tropospheric AMF from version 1.3 to 1.4 to 2.2 over China. Then, we
combine TROPOMI tropospheric $NO_2$ column data and the adjusted tropospheric AMFs by using the differences
among the versions, to correct for the effect of the overestimation of the AMFs used in the previous $NO_2$ version
retrievals. We take the TROPOMI tropospheric AMF version 2.2 data as reference, and calculate the daily ratios
of the AMFs in previous versions and it in the same locations, then the AMFs in these previous versions are (at



least partly) adjusted to avoid the effect of the overestimation. The TROPOMI tropospheric $NO_2$ columns in
these previous versions are thus (partly) corrected by combining the adjusted AMFs and the original observed
slant columns. The result is provided in Fig. 8, with increasing of tropospheric $NO_2$ column of up to 22 % as
compared to the TROPOMI data products.

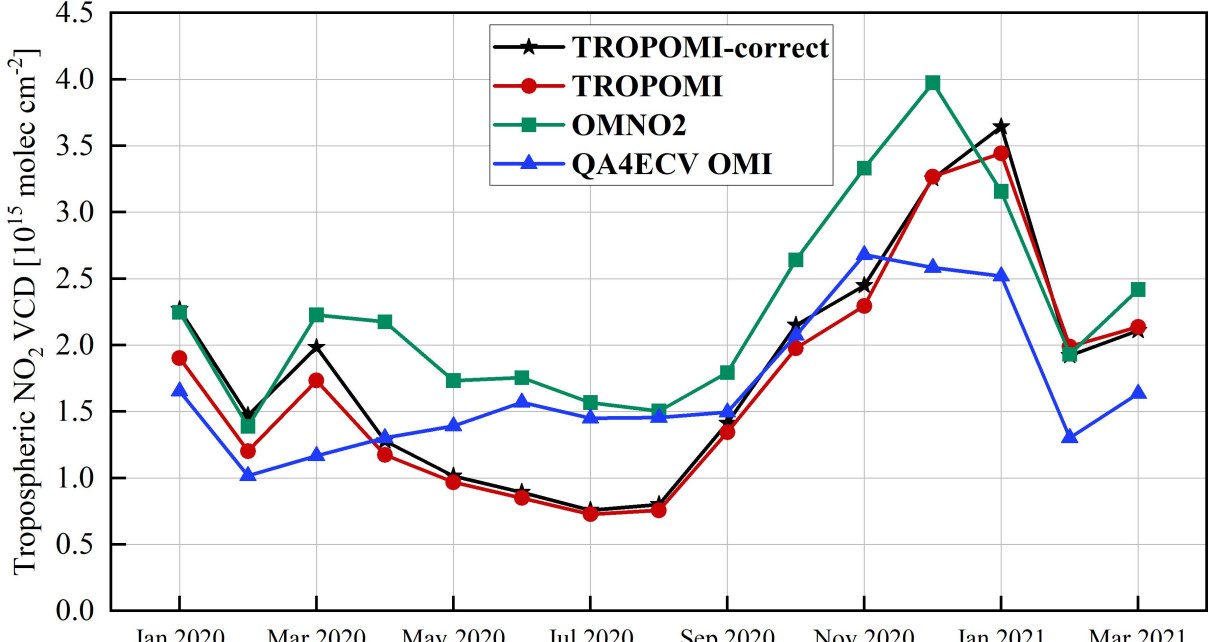

**Figure 8.** Time series of tropospheric $NO_2$ column monthly means retrieved from TROPOMI (red), TROPOMI
with corrected by AMF (black), OMNO2 (green) and QA4ECV OMI (blue) observations over China.

**3.5 $NO_2$ changes during lockdown**
China implemented nationwide restrictions to halt the spread of COVID-19 after the 2020 Spring Festival, such
as implementing strict travel restrictions and suspending factory productions. The nationwide lockdown in China



due to the outbreak of COVID-19 caused large-scale and prolonged shutdowns in rural and urban areas,
consequently, leading to a significant reduction of $NO_2$. In this section we use tropospheric $NO_2$ column data
derived from TROPOMI, OMNO2 and QA4ECV OMI observations to evaluate their sensitivities to detect the
$NO_2$ changes during the COVID-19 lockdown.

We first linearly interpolate the tropospheric $NO_2$ VCDs derived from TROPOMI, OMNO2 and QA4ECV OMI
in February of 2019 and 2021, to create the expected tropospheric $NO_2$ VCDs in February 2020 over China.
Then, we calculate the differences between the observed VCDs and the expected VCDs in February 2020 to
demonstrate the $NO_2$ reduction during the COVID-19 lockdown (Fig. 9). As a result, the reduction during
lockdown derived from TROPOMI ($9.97 \times 10^{14}$ molecules $cm^{-2}$) is significantly greater than from OMNO2 ($5.89$
$\times 10^{14}$ molecules $cm^{-2}$) and QA4ECV OMI ($3.25 \times 10^{14}$ molecules $cm^{-2}$). Moreover, the extent of this $NO_2$
reduction is larger over high pollution regions (e.g. for the BTH region, $3.75 \times 10^{15}$ molecules $cm^{-2}$ in TROPOMI,
$2.75 \times 10^{15}$ molecules $cm^{-2}$ in OMNO2 and $5.85 \times 10^{14}$ molecules $cm^{-2}$ in QA4ECV OMI), due to the stronger
impact of lockdown on these regions with large numbers of industrial facilities and heavy traffic flows. It is
worth noting that the TROPOMI $NO_2$ VCD data in February 2019 using to create the expected $NO_2$ VCD during
lockdown is retrieved in version 1.3, as well as the observed $NO_2$ VCD data in February 2020 using to compare.
But the $NO_2$ VCD data in February 2021, another data source for creating the expected $NO_2$ VCD, is retrieved in
version 1.4. Thus, we use the adjusted TROPOMI tropospheric AMFs as described in Section 3.4, to correct the
TROPOMI tropospheric $NO_2$ VCDs in February of 2019 and 2020 with a low bias. We find that the reduction of
tropospheric $NO_2$ column during lockdown using the adjusted AMFs is $6.66 \times 10^{14}$ molecules $cm^{-2}$ over China,
reflecting that an overestimation of $NO_2$ column reduction during lockdown could be caused by using
TROPOMI data before and after the activation of version 1.4.



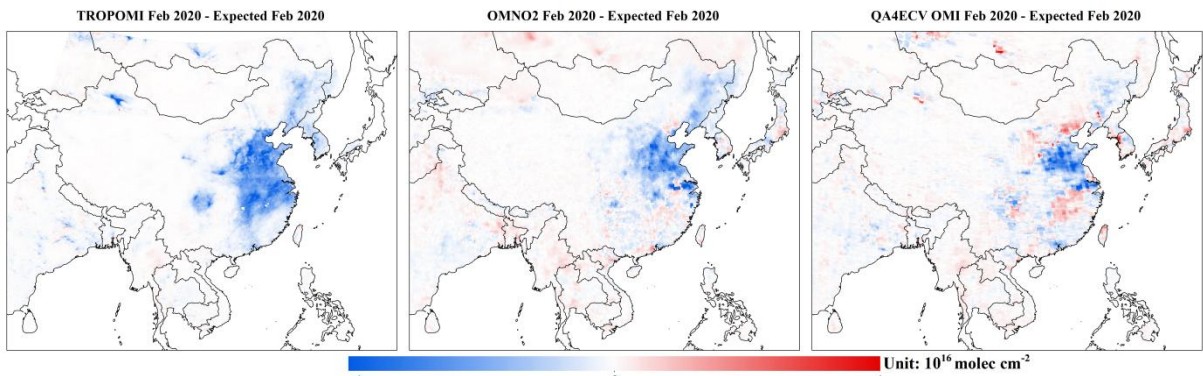


**Figure 9.** Differences between the observed and the expected tropospheric NO$_2$ column derived from TROPOMI

(left panel), OMNO2 (center panel) and QA4ECV OMI (right panel) in February 2020.

## 4 Conclusion

In this work, tropospheric NO$_2$ columns of the TROPOMI version 1.3-2.4 data product are validated using OMI-
derived OMNO2 data and QA4ECV data. The tropospheric column, spatial-temporal distribution, seasonal
variation of the TROPOMI NO$_2$ data in the different versions are presented and compared to the OMNO2 and
QA4ECV OMI NO$_2$ observations over China. In addition, the changes of the TROPOMI AMFs under the
different NO$_2$ retrieval version conditions are measured. The major conclusions are summarized as follows.

(1) The tropospheric NO$_2$ columns derived from TROPOMI version 1.3 data are lower than those derived from
OMNO2 data (54 %) and QA4ECV OMI data (50 %) over China, which mainly due to the overestimation of
cloud pressure retrieved by the FRESCO cloud retrieval algorithm, and subsequently the overestimation of the
AMF for scenes with small cloud fractions. As a consequence, a significant increase by 38 % of tropospheric
NO$_2$ columns, derived with the version 1.4 improved FRESCO-wide cloud retrieval, was identified as compared



with the previous version. Moreover, TROPOMI tropospheric $NO_2$ column in version 2.2 is 14 % higher than in
version 1.4, due to the adjusted surface albedo for cloud-free scenes.

(2) The upgrade to the current TROPOMI $NO_2$ version 2.4 with the DLER surface albedo led to a significant
increase by 3.44 x $10^{14}$ molecules $cm^{-2}$ of tropospheric $NO_2$ columns over vegetation, as compared with the
previous version, due to the strong effect of the DLER over vegetation in the near infrared. Moreover, the results
for tropospheric $NO_2$ seasonal variation by comparison of TROPOMI $NO_2$ version 1.3-2.2 data with OMNO2
data and QA4ECV OMI data are provided. TROPOMI data shows strongest tropospheric $NO_2$ seasonal variation
compared to the other data. Additionally, the changes in the TROPOMI $NO_2$ version 1.3-2.2 retrievals lead to
enhance the seasonal effect of tropospheric $NO_2$, due to the seasonal variation of cloud pressure which is
provided more realistic by the FRESCO-wide cloud algorithm in version 1.4 and the adjusted surface albedo for
cloud-free scenes in version 2.2.

(3) TROPOMI AMF in version 1.4 is lower by 32 % than in version 1.3, mainly due to the implementation of the
FRESCO-wide algorithm. The difference of TROPOMI AMF from version 1.4 to 2.2 is relatively minor, range
between a 5 % overestimation and a 3 % underestimation due to the adjusted surface albedo. The TROPOMI
AMF in version 2.4 using the DLER climatology is 3 % lower than in version 2.2. Overall, the TROPOMI AMF
in version 1.3-2.4 over China is given, and based on it, the effects of the underestimation of TROPOMI
tropospheric $NO_2$ column in the previous version retrievals are (at least partly) addressed. In addition, the
reduction of $NO_2$ column during COVID-19 lockdown using the adjusted TROPOMI AMF is presented, and a
33 % overestimation of $NO_2$ column reduction during lockdown is measured as compared to the TROPOMI $NO_2$
data products, due to using TROPOMI data before and after the activation of the $NO_2$ version 1.4.



*Data availability.*


TROPOMI data are obtained from (https://disc.gsfc.nasa.gov/datasets); OMNO2 data are obtained from
(https://aura.gesdisc.eosdis.nasa.gov/data); QA4ECV OMI data are obtained from (http://www.qa4ecv.eu/ecvs).

*Author contributions.*
**Jianbin Gu:** Conceptualization, Methodology, Writing - original draft. **Jinhua Tao:** Data curation. **Xiaoxia**
**Liang and Shipeng Song:** Validation. **Yanfang Tian and Liangfu Chen:** Writing - review & editing.

*Declaration of competing interest.*
The authors declare that they have no known competing financial interests or personal relationships that could
have appeared to influence the work reported in this paper.

*Acknowledgements.*
We gratefully acknowledge TROPOMI, OMI and QA4ECV science teams for making data publicly available.

*Funding.*
This research did not receive any specific grant from funding agencies in the public, commercial, or not-for-
profit sectors.

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
