# Peer review of "Evaluation of TROPOMI operational standard NO2 column"

_EGUsphere, 2023_

## Author Comment (AC1)

**Response to Reviewers**

**Response to comments from Reviewer #1**

**[Comment 1]** *l 27: "We find a 38 % increase of tropospheric $NO_2$ in version 1.4 due to improved FRESCO-wide cloud retrieval, and a 14 % increase in version 2.2 due to adjusted surface albedo for cloud-free scenes."*

*This result is very similar to the conclusion drawn in paper of van Geffen, 2022, where the retrieval changes in version 2.2 (and 1.4) are discussed. Also the comparison against OMI-QA4ECV are presented in this paper.*

**[Response]** Thanks for your comments. We really appreciate the study on the impact of TROPOMI $NO_2$ v2.2 retrieval improvements on a global scale by van Geffen et al. (2022). In this work we follow previous studies and evaluate the impact of TROPOMI $NO_2$ v1.3-2.4 retrieval improvements over China by using TROPOMI, OMNO2 and QA4ECV OMI data. We have extended the study period to TROPOMI $NO_2$ v2.4, find an increase by 27-40 % of tropospheric $NO_2$ with the introducing of v2.4 over vegetation. Furthermore, we find that TROPOMI v1.3-2.2 data shows strongest tropospheric $NO_2$ seasonal variation compared to OMNO2 and QA4ECV OMI data, and this seasonal effect was enhanced with the tropospheric $NO_2$ retrieval version upgrades. Lastly, we conduct a correction for the underestimation of TROPOMI tropospheric $NO_2$ in the previous version retrievals, and find a 33 % overestimation of $NO_2$ reduction during the COVID-19 lockdown over China when using TROPOMI data before and after the activation of the $NO_2$ version 1.4.

**[Comment 2]** *l 29: "We show that the upgrade to version 2.4 with new DLER surface albedo, led to an increase by $3 \times 10^{14}$ molecules $cm^{-2}$ of tropospheric $NO_2$ over vegetation."*

*The paper includes only two months of data of version 2.4, which I would judge is not enough to document the impact of the upgrade to v2.4. Possible weather influences are not discussed by the authors. The increase of $NO_2$ over vegetation has been*

*discussed in the release documentation, the ATBD and the readme file. As indicated below, I was not convinced by the analysis on this topic.*

[Response] Thanks for the comments. We have extended the TROPOMI NO$_2$ v2.4 data series to December 2022, and add a comparison of daily TROPOMI tropospheric NO$_2$ columns in December of 2020 (v1.4), 2021 (v2.2) and 2022 (v2.4) over Fujian province and China, as well as a discussion on the impacts of TROPOMI NO$_2$ v2.4 retrieval over vegetation in winter month. Therefore, in this manuscript we investigate the impacts of TROPOMI NO$_2$ v2.4 improvements under conditions with lush vegetation (summertime) and withered vegetation (winertime) over high vegetation coverage and the whole China. We find that from v1.4 to 2.2 the TROPOMI tropospheric NO$_2$ enhancement over China (7 %, 2.30 $\times$ 10$^{14}$ molecules cm$^{-2}$) is greater than over Fujian which is the province with the highest vegetation coverage in China (1 %, 0.17 $\times$ 10$^{14}$ molecules cm$^{-2}$) in December, similar as the comparison between them in August. However, from v2.2 to 2.4 the TROPOMI tropospheric NO$_2$ column in December is decreased over the whole China (20 %, 6.87 $\times$ 10$^{14}$ molecules cm$^{-2}$) but increased over Fujian (4 %, 1.14 $\times$ 10$^{14}$ molecules cm$^{-2}$). We infer that the impact of the TROPOMI v2.4 improvements with the DLER surface albedo on NO$_2$ column enhancement is relatively stronger over higher vegetation coverage, under the condition with lush vegetation and low NO$_2$ level in summer. Furthermore, in winter when the condition has been changed with withered vegetation and high NO$_2$ emission, the impact of TROPOMI DLER in NO$_2$ v2.4 retrieval is even more obvious. Please see Line 352-356, Lie 368-380 and Figure 4 in the revised manuscript.

TROPOMI NO$_2$ issue 2.2 product documentation (S5P-MPC-KNMI-PRF-NO$_2$) (released in July 2022) tested the impact of DLER in v2.4 for September 2020 on a global scale, suggested that the impact on tropospheric NO$_2$ over Europe, North America and East China is relatively minor, but is substantial over vegetated regions like South America or Central Africa. Our manuscript provides a validation and extension of the tested result in S5P-MPC-KNMI-PRF-NO$_2$ document, as well as a quantitative measurement of the impact of TROPOMI NO$_2$ v2.4 improvements on the

basis of actual observation data.

**[Comment 3]** *l 31: "we demonstrate that TROPOMI data shows strongest tropospheric NO₂ seasonal variation compared to OMNO2 data and QA4ECV OMI data, and this seasonal effect was enhanced with the tropospheric NO₂ retrieval version upgrades": In the paper by van Geffen it was also reported that the largest increases occur in wintertime, so not really a new results.*

**[Response]** Thanks for your comment. We really appreciate the research on the impact of TROPOMI NO₂ v2.2 retrieval upgrade on a global scale by van Geffen et al. (2022). van Geffen et al. (2022) suggested that TROPOMI tropospheric NO₂ v2.2 data is larger than v1.x data, depending on the level of pollution and season, with the largest impact occurs in wintertime. In this manuscript we make some tentative attempts and research on the impacts of TROPOMI NO₂ v1.3-2.4 retrieval upgrades, we find that TROPOMI v1.3-2.2 data shows strongest seasonal variation of tropospheric NO₂ compared to OMNO2 and QA4ECV OMI data, and the improvements in the TROPOMI NO₂ retrieval upgrades lead to form stronger effect of tropospheric NO₂ seasonal variation.

**[Comment 4]** *l 33: "we arrive at a correction for the underestimation of TROPOMI NO₂ column in previous versions"*
*In this respect the paper is a bit late. Such corrections are no longer relevant given the S5P-PAL (available since december 2021) and the reprocessing of v2.4 (available since March 2023). Such corrections have been introduced before, e.g. Riess et al, 2021.*

**[Response]** Thanks for the comments. Riess et al. (2022) used TROPOMI NO₂ v1.2-2.1 and QA4ECV OMI data, applied an artificial neural network, investigated the impact of the COVID-19 pandemic on ship NO₂ pollution over European seas, and found that NOx emissions from ships reduced by 20-25 % during the pandemic. In this work we use TROPOMI NO₂ v1.3-2.4, OMNO2 and QA4ECV OMI data, derive the changes of TROPOMI tropospheric AMF from v1.3-2.4, investigate the

impact of the COVID-19 lockdown on $NO_2$ column over China, we also obtain the expected $NO_2$ reduction during the lockdown over China retrieved from TROPOMI, OMNO2 and QA4ECV OMI data, and find a 33 % overestimation of $NO_2$ reduction during the lockdown over China when using TROPOMI data before and after the activation of v1.4.

**[Comment 5]** *l 35: "We also find a 33 % overestimation of $NO_2$ reduction during the COVID-19 lockdown over China when using TROPOMI data before and after the activation of the $NO_2$ version 1.4." This follows directly from the large increase introduced in v1.4 and is a bit a trivial result. A better analysis of the COVID period was a main motivation to launch the S5P-PAL reprocessing.*

**[Response]** Thanks for the comments. On the one hand, although the upgrade to TROPOMI v1.4 with the improved FRESCO cloud retrieval can lead to a significant increase of tropospheric $NO_2$ as compared with the previous version. Quantitative measurement the impact of TROPOMI v1.4 retrieval improvements on $NO_2$ column changes during the COVID-19 lockdown over China is valuable for the application of TROPOMI-derived $NO_2$ measurement.

On the other hand, TROPOMI operational, OMNO2 and QA4ECV OMI data, instead of S5P-PAL data, are used to investigate the COVID-19 lockdown on $NO_2$ pollution over China in this manuscript is mainly due to the following reasons. Firstly, the aim of our manuscript is to evaluate the impacts of TROPOMI $NO_2$ v1.3-2.4 retrieval improvements over China, and thus, the S5P-PAL $NO_2$ dataset, which is a reprocessing of the TROPOMI official $NO_2$ data product, is not used in this manuscript as a reference for comparison. We appreciate the contribution of this dataset in supporting research on the impact of the COVID-19 lockdown by satellite-derived $NO_2$ observations, and will concern this dataset in the future work on $NO_2$ monitoring during the lockdown. Secondly, up to date OMI and TROPOMI have been the main data sources in satellite monitoring of $NO_2$ (Biswal et al., 2021), the retrieval of tropospheric $NO_2$ from QA4ECV OMI proceeds along the same lines as from TROPOMI, and is similar in many aspects (Riess et al., 2022). Considering the

QA4ECV OMI NO$_2$ data product is available before 30 March 2021, OMNO2 data is used to compare with TROPOMI v2.2 (from July 2021-June 2022) and v2.4 (from July 2022-) data. Therefore, in our manuscript the comparisons of TROPOMI NO$_2$ v1.3-2.4 data with OMNO2 and QA4ECV OMI NO$_2$ data can already reflect the impacts of TROPOMI NO$_2$ retrieval improvements from v1.3-2.4.

**[Comment 6]** *Why do the authors focus only on China? The author team is from China, but the paper is submitted to an international journal. It would be just as relevant to know the impacts over Europe, USA, the tropics etc. As indicated by Van Geffen et al., 2022, the impact seems to be quite dependent on the region.*

**[Response]** Thanks for the comments. In this manuscript we focus on investigating the impacts of TROPOMI NO$_2$ v1.3-2.4 improvements on NO$_2$ column changes over China. We select China as the study area because that China is one of the regions with heaviest NO$_2$ pollution in the world, monitoring NO$_2$ columns over China can well demonstrate its spatial-temporal change characteristics, and furthermore, quantitative measurement of NO$_2$ column changes in TROPOMI different version periods over China can clearly reflect the impacts of these version improvements on NO$_2$ retrieval.

**[Comment 7]** *Why is a reference to the S5P-PAL dataset (https://data-portal.s5p-pal.com/products/no2.html) not included? This PAL dataset was generated to remove the jumps between versions, e.g. for Covid studies.*

**[Response]** Thanks for your comment. Firstly, OMI and TROPOMI have been the main data sources in satellite monitoring of NO$_2$ to date, Retrieval of tropospheric NO$_2$ from QA4ECV OMI proceeds along the same lines as from TROPOMI. Thus QA4ECV OMI data is widely used as reference in previous studies on investigations of impacts of TROPOMI NO$_2$ version upgrades (Riess et al., 2022, van Geffen et al., 2022). Secondly, since the QA4ECV OMI NO$_2$ data is available before 30 March 2021, in order to investigate the impacts of TROPOMI v2.2 (from July 2021-June 2022) and v2.4 (from July 2022-) on NO$_2$ retrieval, OMNO2 data is used to compare. Lastly, the S5P-PAL NO$_2$ dataset is a reprocessing of the TROPOMI official NO$_2$ data

product, while our work aims to evaluate the impacts of TROPOMI $NO_2$ v1.3-2.4 retrieval improvements over China, thus S5P-PAL $NO_2$ data is not applied to compare with TROPOMI operational $NO_2$ data. Overall, comparisons of TROPOMI $NO_2$ v1.3-2.4 data with OMNO2 and QA4ECV OMI data, instead of S5P-PAL data, have been able to well accomplish the evaluation of the impacts of these different retrieval version improvements.

**[Comment 8]** *There is a new reprocessing available since March 2023, covering the full mission duration, see*

*https://sentinels.copernicus.eu/web/sentinel/-/copernicus-sentinel-5-precursor-full-mission-reprocessed-datasets-further-products-release.*

*The seasonality linked to the DLER albedo update can be studied with this new dataset. I realise that the paper was written before the reanalysis became available, so this is not a main reason for my negative judgement.*

**[Response]** Thanks for your comments. Copernicus Sentinel-5 Precursor full mission reprocessed datasets have not released when we completed this manuscript. We will concern these new datasets in the future work of air pollution monitoring.

**[Comment 9]** *The paper focusses on China, but the POMINO product is not mentioned at all*

*(http://www.pku-atmos-acm.org/acmProduct.php/#TROPOMI). This is a clear omission.*

**[Response]** Thanks for your comment. OMI and TROPOMI have been the main data sources in satellite monitoring of $NO_2$ to date, because the retrieval of tropospheric $NO_2$ from QA4ECV OMI proceeds along the same lines as from TROPOMI, thus QA4ECV OMI data is used in this work on investigations of impacts of TROPOMI $NO_2$ version upgrades. Moreover, the QA4ECV OMI $NO_2$ data product is available before 30 March 2021, so OMNO2 data is also used to compare with TROPOMI data in the v2.2 (from July 2021-June 2022) and v2.4 (from July 2022-) periods. Additionally, the POMINO-TROPOMI product is retrieved from the TROPOMI

instrument and based on calculation of tropospheric AMFs by applying a radiative transfer model LIDORT. But in this work we focus on evaluating TROPOMI's capability to detect tropospheric $NO_2$ in different version retrievals itself, and thus the POMINO-TROPOMI product, which is also derived by TROPOMI, is not applied in this manuscript. We appreciate the contribution of this dataset in satellite-derived $NO_2$ monitoring for China, and will concern this dataset in the future work of $NO_2$ pollution monitoring in China.

**[Comment 10]** *It was surprising that Lamsal et al, https://doi.org/10.5194/amt-14-455-2021, is not cited for v4 of OMNO2A.*

**[Response]** Done as suggested. We have reviewed and cited the related findings in the paper of Lamsal et al. (2021) accordingly. Please see Line 206 in the revised manuscript.

**[Comment 11]** *There is no reference to the product readme file and user manual documents of TROPOMI $NO_2$: these documents inform users of the updates of the processor and main impacts and are therefore relevant for this paper.*

**[Response]** Thanks for your comment. The product readme file and user manual document of TROPOMI $NO_2$ (e.g. S5P-MPC-KNMI-PRF-$NO_2$ issue 2.2 document) is already cited in our manuscript. Please see Line 170-172, Line 293-295, and Line 322-324 in our manuscript.

**[Comment 12]** *As indicated by the authors, the v2.4 upgrade is "not well documented" in the peer-reviewed literature. I can sympathise with this statement. But this version is new, and is also used in the recent reprocessing. Did the authors get in contact with the retrieval team, which would be a normal step to take?*

**[Response]** Thanks for your comment. We have dug up the information on TROPOMI $NO_2$ v2.4 retrieval improvements from official ESA channels, including TROPOMI ATBD tropospheric and total $NO_2$ (S5P-KNMI-L2-0005-RP) issue 2.4 document, S5P-MPC-KNMI-PRF-$NO_2$ issue 2.2 document, and some other relevant

papers (e.g. Tilstra et al., 2017).

**[Comment 13]** *The analyses presented by the authors are rather straightforward, consisting of simple comparisons of averages of the tropospheric column and the tropospheric air-mass factor between different retrieval versions. A more in-depth analysis of the differences is missing, and the general conclusions broadly agree with what has already been reported before.*

**[Response]** Thanks for your comments. In this work we evaluate the impacts of TROPOMI $NO_2$ v1.3-2.4 retrieval upgrades over China, quantitative measure these impacts on $NO_2$ column changes, and also make deep analysis and discussions on these impacts on $NO_2$ retrievals. Take the impact of TROPOMI $NO_2$ v2.4 retrieval upgrade as an example, the DLER surface albedo using in TROPOMI v2.4 accounts for the directionality or viewing-angle dependence of the scattering at the surface, especially over vegetation in the near infrared. Thus according to this strong effect of the DLER over vegetation, we evaluate to the new DLER surface albedo in and its impact on the TROPOMI $NO_2$ columns, to better understand the detection of $NO_2$ under condition of vegetation coverage. Moreover, considering weather variability may play a big role in vegetation coverage changes, we compare the TROPOMI tropospheric $NO_2$ VCD daily data in August (condition with lush vegetation) and December (condition with withered vegetation) of 2020 (v1.3, 1.4), 2021 (v2.2) and 2022 (v2.4) over Fujian province (the province with the highest vegetation coverage in China), as well as over China. We find that from v1.3 to 2.2 to 2.4, over China the August TROPOMI tropospheric $NO_2$ column is increased by 9 % (1.06 $\times$ $10^{14}$ molecules $cm^{-2}$) and 5 % (0.73 $\times$ $10^{14}$ molecules $cm^{-2}$) respectively, and in comparison, the increase in August TROPOMI tropospheric $NO_2$ column over Fujian from v1.3 to 2.2 is relatively minor (2 %, 0.35 $\times$ $10^{14}$ molecules $cm^{-2}$), but the tropospheric $NO_2$ enhancements over this region from v2.2 to 2.4 are presented, with a substantial increase (16 %, 2.42 $\times$ $10^{14}$ molecules $cm^{-2}$). The impact of the TROPOMI v2.4 improvements with the DLER surface albedo on $NO_2$ column enhancement is relatively stronger over higher vegetation coverage, under the

condition with lush vegetation and low $NO_2$ level in summer. Furthermore, in winter when the condition has been changed with withered vegetation and high $NO_2$ emission, the impact of TROPOMI DLER in $NO_2$ v2.4 retrieval is even more obvious. For instance, from v1.4 to 2.2, the TROPOMI tropospheric $NO_2$ enhancement over China (7 %, 2.30 $\times$ $10^{14}$ molecules cm$^{-2}$) is greater than over Fujian (1 %, 0.17 $\times$ $10^{14}$ molecules cm$^{-2}$) in December, similar as the comparison between them in August. However, from v2.2 to 2.4 the TROPOMI tropospheric $NO_2$ column in December is decreased over the whole China (20 %, 6.87 $\times$ $10^{14}$ molecules cm$^{-2}$), but increased over Fujian (4 %, 1.14 $\times$ $10^{14}$ molecules cm$^{-2}$). Please see Section 3.2 in the revised manuscript.

**[Comment 14]** *The data series stops in September 2022. This is a very short period (only two months) to make any statements on the version 2.4 data. Because the albedo climatology is available on a monthly basis, it would be important to document a full year of data.*

**[Response]** Thanks for your comment. We have extended the data series to December 2022, and add comparisons of TROPOMI $NO_2$ and OMI $NO_2$ in December of 2020-2022 over Fujian province and China, as well as a discussion on the impacts of TROPOMI $NO_2$ v2.4 improvements during withered month. Thus, the impacts of TROPOMI $NO_2$ v2.4 improvements under conditions with lush vegetation (summertime) and withered vegetation (winertime) over high vegetation coverage and the whole China are all investigated. Please see Line 364-380 and Figure 4 in the revised manuscript. In addition, since TROPOMI $NO_2$ v2.4 data is in operation from July 2022, a full year of this data can not be obtained to date.

**[Comment 15]** *I found the analysis of the impact over vegetation not convincing: How can we compare relative differences in Fujian and the entire China? Many aspects may play a role here. Furthermore, the analysis is limited to one month, which is not convincing as weather variability may play a big role. So I do not think the authors have presented enough evidence to quantify the increase due to v2.4.*

**[Response]** Thanks for your comments. On the one hand, because the DLER surface albedo using in TROPOMI v2.4 accounts for the directionality or viewing-angle dependence of the scattering at the surface, especially over vegetation. We select Fujian province which is the province with the highest vegetation coverage in China as the study area to investigate the strong effect of the DLER over vegetation, and here we also measure the impact of the DLER in TROPOMI v2.4 on $NO_2$ retrieval over the whole China as a reference. On the other hand, we have extended the data series to December 2022, and add comparisons of TROPOMI $NO_2$ and OMI $NO_2$ in December of 2020-2022 over Fujian province and China, thus the impacts of TROPOMI $NO_2$ v2.4 improvements under conditions with lush vegetation and withered vegetation are both investigated, and the relative differences of TROPOMI $NO_2$ column changes due to the introduction of v2.4 over Fujian and the entire China are derived. Please see Line 364-380 and Figure 4 in the revised manuscript.

**[Comment 16]** *Concerning the seasonality, section 3.3: I could not reconcile the results of Figure 5 with figure 6, for QA4ECV OMI. Is there a mistake in figure 5, since the seasonality seems much too small?*

**[Response]** Thanks for your comments. Sorry for the confusion caused by the unclear expressions of the figure captions of Figure 5 and 6. We have revised these figure captions in the revised manuscript. In our manuscript Figure 5 shows the result of $NO_2$ seasonal variation for the whole China retrieved by TROPOMI, OMNO2 and QA4ECV OMI data. From Figure 5 it can be clearly seen that over China compared to OMNO2 and QA4ECV OMI data, TROPOMI data shows strongest seasonal variation of tropospheric $NO_2$ columns, and the extents of the observed $NO_2$ changes in winter or summer month retrieved from TROPOMI exceed those retrieved from OMI. In addition, although QA4ECV OMI follows a more similar $NO_2$ retrieval algorithm to TROPOMI relative to OMNO2, the increase in winter and decrease in summer of $NO_2$ observed with QA4ECV OMI (-0.5 % and -5 %) are even smaller than those observed with OMNO2 (4 % and -18 %) over China. Taking this into account, we conclude that the FRESCO-wide cloud algorithm using in the $NO_2$ retrieval has a

positive impact on the clear demonstration of seasonal variation of TROPOMI tropospheric $NO_2$.

Figure 6 demonstrate the result of $NO_2$ time series for the Beijing-Tianjin-Hebei (BTH), Yangtze River Delta (YRD) and Pearl River Delta (PRD) region in China by using TROPOMI, OMNO2 and QA4ECV OMI data. From Figure 6 it can be clearly seen that over these three regions with high $NO_2$ pollution, The characteristics of $NO_2$ seasonal variation derived from TROPOMI, OMNO2 and QA4ECV OMI data are all significant. We calculate the differences between TROPOMI and QA4ECV OMI tropospheric $NO_2$ daily VCDs for each selected pollution region and each month from November 2019 to November 2020, we find that compared to the QA4ECV OMI $NO_2$ retrieval, the FRESCO cloud algorithm using in the TROPOMI $NO_2$ retrieval has a strongly positive impact on the tropospheric $NO_2$ seasonal cycle, especially in high pollution regions.

**Reference**

Biswal, A., Singh, V., Singh, S., Kesarkar, A. P., Ravindra, K., Sokhi, R. S., Chipperfield, M. P., Dhomse, S. S., Pope, R. J., Singh, T. and Mor, S.: COVID-19 lockdown-induced changes in $NO_2$ levels across India observed by multi-satellite and surface observations, Atmos. Chem. Phys., 21(6), 5235–5251, doi:10.5194/acp-21-5235-2021, 2021.

Lamsal, L., A. Krotkov, N., Vasilkov, A., Marchenko, S., Qin, W., Fasnacht, Z., Joiner, J., Choi, S., Haffner, D., H. Swartz, W., Fisher, B. and Bucsela, E.: Ozone Monitoring Instrument (OMI) Aura nitrogen dioxide standard product version 4.0 with improved surface and cloud treatments, Atmos. Meas. Tech., 14(1), 455–479, doi:10.5194/amt-14-455-2021, 2021.

Riess, T. C. V. W., Boersma, K. F., Van Vliet, J., Peters, W., Sneep, M., Eskes, H. and Van Geffen, J.: Improved monitoring of shipping $NO_2$ with TROPOMI: Decreasing NOx emissions in European seas during the COVID-19 pandemic, Atmos. Meas. Tech., 15(5), 1415–1438, doi:10.5194/amt-15-1415-2022, 2022.

Tilstra, L. G., Tuinder, O. N. E., Wang, P. and Stammes, P.: Surface reflectivity climatologies from UV to NIR determined fromEarth observations by GOME-2 and SCIAMACHY, J.

Geophys. Res., 122(7), 4084–4111, doi:10.1002/2016JD025940, 2017.

Van Geffen, J., Eskes, H., Compernolle, S., Pinardi, G., Verhoelst, T., Lambert, J. C., Sneep, M., Linden, M. Ter, Ludewig, A., Folkert Boersma, K. and Pepijn Veefkind, J.: Sentinel-5P TROPOMI $NO_2$ retrieval: impact of version v2.2 improvements and comparisons with OMI and ground-based data, Atmos. Meas. Tech., 15(7), 2037–2060, doi:10.5194/amt-15-2037-2022, 2022.

---

## Author Comment (AC2)

**Response to Reviewers**

**Response to comments from Reviewer #2**

[**Comment 1**] *Line 123-124, the definition of SCD and VCD need be rephrased since they are not accurate in current description. Please refer to the product documentation or the DOAS book (Platt and Stutz, 2008).*

[**Response**] Thanks for the comment and suggestion. We have reviewed the DOAS book (Platt and Stutz, 2008) and the TROPOMI $NO_2$ issue 2.2 product documentation (S5P-MPC-KNMI-PRF-$NO_2$), OMNO2 version 4.0 product documentation, QA4ECV $NO_2$ version 1.1 product documentation. In the book and product documentations, the definition of SCD and VCD are all "slant column density" and "vertical column density" (e.g. Page 345, 347 in the book), the same as mentioned in our manuscript.

[**Comment 2**] *In Fig.2, in addition to the version of retrievals, the periods are also different. How can we attribute the differences to the retrieval itself rather than the differences of $NO_2$ in temporal? Please clarify and provide the evidence.*

[**Response**] Thanks for your comments. Figure 2 shows the differences in monthly mean tropospheric $NO_2$ columns derived from TROPOMI data and QA4ECV OMI data between December 2019-March 2020 and December 2020-March 2021, which is aim to present the differences between TROPOMI $NO_2$ v1.3 and 1.4 by QA4ECV OMI $NO_2$ data as reference on a monthly average basis, not for demonstrating the differences of $NO_2$ in temporal by these datasets themselves.

The differences of $NO_2$ in temporal retrieved by TROPOMI, OMNO2 and QA4ECV OMI data are discussed in section 3.1, 3.3 and Figure 1, 5, 6 in our manuscript. We find that TROPOMI v1.3 tropospheric $NO_2$ VCDs have the largest decrease in the summer months (e.g. 52 % for June, 54 % for July and 50 % for August), and the smallest decrease in the winter months (e.g. 15 % for January, 13 % for February and 22 % for March), as compared to the OMNO2 tropospheric VCDs. Similar seasonal differences exist in the comparison of the TROPOMI tropospheric $NO_2$ VCDs to the

QA4ECV OMI tropospheric $NO_2$ VCDs (e.g. -46 % for June, -50 % for July, -48 % for August and 15 % for January, 18 % for February, 49 % for March). Furthermore, TROPOMI v1.3-2.2 data shows strongest seasonal variation of tropospheric $NO_2$ columns compared to OMNO2 and QA4ECV OMI data, the extents of the observed $NO_2$ changes in winter or summer month retrieved from TROPOMI exceed those retrieved from OMI.

**[Comment 3]** *Sect 3.2, when the authors discuss the impacts of DLER over vegetation, only a summer month (August) were selected for Fujian province and China. I guess this month was chose to represent the condition with vivial vegetation. However, a comparative withered season/month should also be considered to show the change in surface albedo and further impacts in DLER and $NO_2$ products.*

**[Response]** Thanks for the comment and suggestion. We have added a comparison of daily TROPOMI tropospheric $NO_2$ columns in December of 2020 (v1.4), 2021 (v2.2) and 2022 (v2.4) over Fujian province and China, as well as a discussion on the impacts of DLER in TROPOMI $NO_2$ v2.4 retrieval over vegetation during withered month. The impacts of TROPOMI $NO_2$ v2.4 improvements under conditions with lush vegetation (summertime) and withered vegetation (wintertime) over high vegetation coverage and the whole China are all investigated. The analysis and discussion on the impacts of TROPOMI $NO_2$ v2.4 retrieval under these conditions are given, please see Line 364-380 and Figure 4 in the revised manuscript.

**[Comment 4]** *Fig. 5 and other similar inferred conclusions, I think that an independent ground-based measurements of $NO_2$ VCD datasets can strongly enhance the evidences. Otherwise, it's hard to exclude the upward trends in winter and downward in summer from the seasonal pattern difference from year to year. Similarly, there also other conclusions are not solid and convincible.*

**[Response]** Thanks for the comment and suggestion. In this work we use OMNO2 and QA4ECV OMI $NO_2$ products as reference for comparison, not ground-based $NO_2$ observations, due to the following two reasons. Firstly, up to date systematic and

consistent ground-based $NO_2$ observation data has been only provided till November 2017 (e.g. QA4ECV MAX-DOAS datasets). Secondly, our manuscript focuses on evaluating of TROPOMI $NO_2$ v1.3-2.4 retrieval improvements over China, based on 3-year data with large spatial scale. For instance, Figure 5 covers a period from December 2019 to June 2022. Such a long-term independent ground-based $NO_2$ VCD dataset is difficult to obtain. Moreover, up to date OMI and TROPOMI have been the main data sources in satellite monitoring of $NO_2$ (Biswal et al., 2021), the retrieval of tropospheric $NO_2$ from QA4ECV OMI proceeds along the same lines as from TROPOMI, and is thus similar in many aspects. Thus QA4ECV OMI data is widely used as reference in previous studies on investigations of impacts of TROPOMI $NO_2$ version upgrades (Riess et al., 2022, van Geffen et al., 2022).

In addition, since the QA4ECV OMI $NO_2$ data product is available before 30 March 2021, in order to investigate the impacts of TROPOMI v2.2 (from July 2021-June 2022) and v2.4 (from July 2022-) on $NO_2$ retrieval, in this work OMNO2 data is also used to compare with TROPOMI data. Overall, comparisons of TROPOMI $NO_2$ v1.3-2.4 data with OMNO2 and QA4ECV OMI data have been able to well accomplish the evaluation of the impacts of these different retrieval version improvements.

**[Comment 5]** *Line 426-428, how to create the AMF dataset? By RTM? If it is, please describe the simulation and key inputs in details. And how about the authors' simulation compared to the AMFs used in products retrieval? If not, how to get the AMF?*

**[Response]** Thanks for the comments. The $NO_2$ AMF dataset is created using the Doubling-Adding KNMI radiative transfer model, and the input parameters to the $NO_2$ AMF calculation are surface albedo climatology, priori $NO_2$ profiles, viewing geometry, terrain height and cloud parameters. Please see Line 136-141 in the revised manuscript.

In this work we create the adjusted AMFs by deriving the differences of TROPOMI $NO_2$ AMF from v1.3 to 2.4, please see Table 2 in the revised manuscript. Furthermore,

the comparisons between our simulation (TROPOMI with corrected by AMF) and satellite NO$_2$ data products are conducted. Please see Figure 8 in the revised manuscript.

**[Comment 6]** *Line 487-503, considering there were many literatures that reported the changes of NO$_2$ VCDs during the 2020 lockdown in China (in both spaced-based sensors and ground based MAX-DOAS), the authors could refer to the reported decreases and compared with the expectation in Figure 9.*

**[Response]** Thanks for your suggestion. We have reviewed and cited the related findings in the paper of Ding et al. (2020) as follows: "The expected TROPOMI NO$_2$ reduction over the BTH region (44 %) during the lockdown in February 2020 is consist with the previous study by Ding et al. (2020) who found that most Chinese cities showed strong NO$_2$ emission reductions of 20-50 % in the same period". Please see Line 514-516 in the revised manuscript.

**[Comment 7]** *Better to cite the full name of some nouns for the first time even in the abstract, e.g. Tropomi, OMI, QA4ECV, DLER, etc.*

**[Response]** Done as suggested.

**[Comment 8]** *Line 165, NO$_2$ subscript.*

**[Response]** Done as suggested.

**[Comment 9]** *Line 193, should be "tropospheric and stratospheric NO$_2$ SCDs"*

**[Response]** Done as suggested.

**[Comment 10]** *I would like to suggest to show the monthly series of different products of NO$_2$ VCDs from OMI and TROPOMI in another panel in Figure 1 too, which is helpful to show the absolute differences.*

**[Response]** Done as suggested. Please see Figure 1 in the revised manuscript.

**[Comment 11]** *Line 328-330, the comparison of spatial distribution between annual averages of these three products may be also presented in Fig. 3.*

**[Response]** Done as suggested. Please see Figure 3 in the revised manuscript.

**Reference**

Biswal, A., Singh, V., Singh, S., Kesarkar, A. P., Ravindra, K., Sokhi, R. S., Chipperfield, M. P., Dhomse, S. S., Pope, R. J., Singh, T. and Mor, S.: COVID-19 lockdown-induced changes in $NO_2$ levels across India observed by multi-satellite and surface observations, Atmos. Chem. Phys., 21(6), 5235–5251, doi:10.5194/acp-21-5235-2021, 2021.

Ding, J., van der A, R. J., Eskes, H. J., Mijling, B., Stavrakou, T., van Geffen, J. H. G. M. and Veefkind, J. P.: NOx Emissions Reduction and Rebound in China Due to the COVID-19 Crisis, Geophys. Res. Lett., 47(19), 1–9, doi:10.1029/2020GL089912, 2020.

Platt, U. and Stutz, J.: Differential Optical Absorption spectroscopy, Principles and Applications, ISBN 978-3-540-75776-4, 2008.

Riess, T. C. V. W., Boersma, K. F., Van Vliet, J., Peters, W., Sneep, M., Eskes, H. and Van Geffen, J.: Improved monitoring of shipping $NO_2$ with TROPOMI: Decreasing NOx emissions in European seas during the COVID-19 pandemic, Atmos. Meas. Tech., 15(5), 1415–1438, doi:10.5194/amt-15-1415-2022, 2022.

Van Geffen, J., Eskes, H., Compernolle, S., Pinardi, G., Verhoelst, T., Lambert, J. C., Sneep, M., Linden, M. Ter, Ludewig, A., Folkert Boersma, K. and Pepijn Veefkind, J.: Sentinel-5P TROPOMI $NO_2$ retrieval: impact of version v2.2 improvements and comparisons with OMI and ground-based data, Atmos. Meas. Tech., 15(7), 2037–2060, doi:10.5194/amt-15-2037-2022, 2022.